# Can the New Subway Line Openings Mitigate PM10 Concentration? Evidence from Chinese Cities Based on the PSM-DID Method

**DOI:** 10.3390/ijerph17134638

**Published:** 2020-06-27

**Authors:** Ying Wang, Jing Tao, Rong Wang, Chuanmin Mi

**Affiliations:** 1College of Economics and Management, Nanjing University of Aeronautics and Astronautics, Nanjing 211106, China; yingwang@nuaa.edu.cn (Y.W.); cmmi@nuaa.edu.cn (C.M.); 2School of Economics and Management, Nanjing Institute of Technology, Nanjing 211167, China; wxr920@163.com

**Keywords:** subway, PM10 pollution, PSM–DID method, traffic congestion

## Abstract

The large-scale construction of subway systems, which is viewed as one of the potential measures to mitigate traffic congestion and its resulting air pollution and health impact, is taking place in major cities throughout China. However, the literature on the impact of the new subway line openings on particulate matter with a diameter less than 10 µm (PM10) at the city level is scarce. Employing the Propensity Score Matching–Difference-in-differences method, this paper examines the effect of the new subway line openings on air quality in terms of PM10 in China, using the daily PM10 concentration data from January 2014 to December 2017. Our finding shows that the short-term treatment effect on PM10 is more controversial. Furthermore, for different time windows, the result confirms an increase in PM10 pollution during the short term, while the subway line openings improve air quality in the longer term. In addition, we find that the treatment effect results in high PM10 pollution for cities with 1–2 million people, while it improves air quality for cities with over 2 million people. Moreover, for cities with varying levels of GDP, there is evidence of a reduction in PM10 after the subway line openings. Mechanism analysis supports the conclusion that the PM10 reduction originated from substituting the subway for driving.

## 1. Introduction

The primary air pollutant in terms of PM10 (particulate matter with a diameter less than 10 µm) in China has begun to have adverse effects on the climate [1], visibility [2] and human health [3]. Recent research indicated that short- or long-term exposure to PM10 pollution can induce adverse outcomes for human life [4], reproductive health [5], pregnant women [6], and even infant birthweight [7].

PM10 pollution is produced by various sources, such as industrial exhaust and transport-related emissions in urban areas [8]. Among these sources, long-distance transportation has become an important factor that influences air pollution in urban areas [9]. Therefore, in order to effectively solve the problem of particulate matter (PM) pollution, the Chinese government has taken various measures, such as driving the restriction policy, to control the traffic congestion caused by increasing vehicle ownership and the resulting air pollution in the urban area. In addition to these demand-side strategies, China has also been significantly investing in public transportation infrastructure, such as subway systems. The large-scale construction of subway systems is taking place in major cities throughout China. At the end of 2018, 35 cities in China (excluding Hong Kong, Macao and Taiwan) had subway systems with 185 operating lines, and a total length of 5761.4 km.

Despite the massive investment in subway infrastructure in China, estimations of the impact of the new subway line openings on PM10 pollution are lacking. This paper explores the effect of the new subway line openings on air quality in terms of PM10 concentration, using the Propensity Score Matching–Difference-in-differences (hereinafter abbreviated as “PSM-DID”) method during the period of January 2014 to December 2017. The new subway line openings could result in the traffic diversion effect, which could influence PM10. The traffic diversion effect highlights the reduction of traffic congestion, and thus the improvement of air quality in the urban area [10]. However, Zheng et al. [11] argued that the mitigation of air pollution does not only depend on the traffic diversion force, but also on the pollutant emissions force caused by public transit. Several recent studies (i.e., [12,13,14]) indicated that indoor PM pollution caused by subway lines is higher than outdoor pollution caused by road traffic-generated particles. Indoor sources can also contribute extremely high levels of outside air pollution in heavily populated locations [15]. Therefore, the net effect on PM10 concentration after the subway openings is determined by these two countervailing forces: the subway PM effect and the traffic diversion effect. Thus the treatment effect is still an empirical question.

Our empirical analysis uses the rich spatial and temporal variation in PM10 pollution, and the new subway openings across Chinese cities, during the data period of January 2014 to December 2017. During this period, 12 cities opened their first newly-built subway lines. However, due to several cities opening their second subway line in a shorter time (e.g., no more than 1 year), which may lead to the interaction effects caused by subway line openings that opened at different times [11], we finally selected 7 Chinese cities as the processing group, to estimate the effect of the subway supply on PM10 pollution. The primary empirical strategy explores how PM10 pollution within cities is affected before and after the new subway openings, over time and across space in China. The main identification concern stems from the sample selection bias and individual heterogeneity [16]. For example, the Chinese cities that constructed subway lines were not random, as this related to the level of economic development and the environmental factors of those cities. The sample selection bias and individual heterogeneity may lead to the “differential deviation” of the DID method.

To overcome this concern, we use the PSM-DID method to assess the “net effect” on PM10 concentration before and after the new subway openings in China, by eliminating the influence of heterogeneity and time variation factors [17]. We first use the PSM approach to match the processing group and the control group based on control variables. We define the cities with subway line openings as the processing group, and the cities without any subway line openings as the control group. Then, this paper uses the DID method to estimate the effect of new subway openings on PM10 pollution from January 2014 to December 2017. Secondly, between the processing group and the control group, we focus on changes in PM10 pollution 90 days, 350 days and 500 days before and after the openings of a subway line. This focus on both shorter and longer time windows can better address the concern of un-observables, and it is able to examine both short- and longer-term impacts. Thirdly, we examine the relationship between the new subway line openings and PM10 pollution for three sub-samples of cities with varying population sizes. We divide the sample into three groups: cities with 1–2 million people; cities with 2–3 million people; and cities with 3–4 million people. Similarly, we estimate the mitigation of PM10 pollution after the new subway line openings for cities with varying levels of economic development.

Furthermore, having found that the subway line openings lead to high levels of PM10 pollution in the short term, while it improves air quality in the longer term, we then investigate the influence mechanism that the subway can provide, via its substitution for driving, and the alleviation of traffic congestion. To do so, we examine evidence for heterogeneous responses to the openings of the subway line through one main tailpipe emission, resulting from on-road traffic: CO. Our findings show that the reduction in CO is significantly low during the short term, but it is higher when we increase time windows to 350 days and longer. The findings support the substitution of driving as the mechanism that leads to improved air quality in terms of PM10 in the longer run, but it is noteworthy that this improvement may be small due to the subway-induced PM pollution effect. We also show the robustness of our findings, by choosing PM2.5 as the robustness pollutant.

## 2. Literature Review

### 2.1. The Impact of Air Pollution on Human Health

Particulate matter (PM) is one of the major causes of mortality and negative health effects. For example, the World Health Organization recently estimated that worldwide air pollution exposure contributed to about 4.2 million of the deaths [18]. The latest research of the Global Burden of Diseases ranked PM as the sixth key cause of mortality, being responsible for over 4 million deaths in 2016 [19]. The mortality caused by air pollution was reduced by 15% when the annual PM10 concentration was reduced from 70 to 20 mg/m^3^ [3]. In addition to mortality caused by PM, it was also considered to be an important cause for human respiratory problems (including asthma exacerbation and chronic obstructive pulmonary disease) [20], cardiovascular system diseases (such as an irregular heartbeat and vascular dysfunction) [21], and even lung cancer [22]. Based on the above considerations, it is important and necessary to explore measures in order to solve the pollution problems related to PM10, thus improving air quality and maintaining human health [23,24].

### 2.2. The Sources of Air Pollution

As one of the important factors affecting air quality, much research has tried to investigate the source of PM for improving air quality. Even though the occurrence of particulate pollution is the result of complex processes, as yet it also involves some direct emissions, such as soli dust, automobile exhaust, industrial exhaust, metal processing and secondary ions [25]. In these direct sources, long-distance transportation has become a key factor affecting air pollution in urban areas [26,27,28]. For example, it was reported that PM10 pollution caused by transportation in 2010 ranked as the sixth main pollutant in China, which contributed to around 16,615 Gg of emissions [29], and the car emissions contributed about 63% of CO, 37% of NOx, and more than 20% of PM2.5 in urban areas in China [11].

### 2.3. The Effect of Driving Restriction Policy on Air Pollution

The previous stuies have paid close attention to the effect of driving restrictions on air pollution, in which the results are often controversial. On the one hand, Viard [30] finds that traffic restriction results in a 7% decline in air pollution during one day per week restrictions in Beijing. Similarly, Chen and Jin [31] explore the effectiveness of different environmental measures adopted by the Chinese government to prepare for the 2008 Olympic Games. On the other hand, Davis [32] investigates the effectiveness of driving restrictions in Mexico City, and finds that the driving restriction leads to low air quality due to more households buying a second vehicle. This finding is consistent with the results of Zhang et al. [33], who examine the effect of driving restriction policies implemented in Bogota.

Meanwhile, a new wave of empirical research has recently tried to estimate the relationship between public transit and air pollution [34,35]. Many studies support the conclusion that public transit supply leads to the mitigation of typical tailpipe pollution. For example, using the regression discontinuity (RD) approach, Chen and Whalley [36] first investigate the impact of a new subway opening on air pollution. The results of this paper indicate that the opening of a subway line in Taipei caused a 5–15% reduction in air pollution in terms of carbon monoxide (CO), but little effect on particulates. Similarly, Zheng et al. [11] find an 18% reduction in CO after the opening of a subway line in Changsha, based on the DID method. Gendron-Carrier et al. [37] find that particulate concentrations drop by 4% in a 10-km radius disk surrounding a city center following a subway system opening. Li et al. [10] also find that a subway opening improves air quality (i.e., Air Quality Index), using a distance-based DID method. However, unlike the above studies, using the case of the U.S., Beaudoin et al. [38] find no evidence that an increase in public transit supply reduces air pollution in terms of some pollutant concentrations (e.g., PM). In addition, Bauernschuster et al. [39] find that transit strikes increase air pollution in terms of particulates concentration.

### 2.4. The Influence Mechanism of the Effect of Subway Supply on Air Pollution

Undoubtedly, the mitigation of typical tailpipe pollution (e.g., CO and NO_2_) after a subway line opening is not surprising, because the mitigation of transport-related pollutants is determined by the traffic diversion force. The traffic diversion effect stresses that a new subway line opening could lead some commuting people who relied on private automobiles before to switch to using subway lines [40]. It would reduce traffic congestion and thus improve air quality. In empirical studies, they usually test the mechanism by estimating heterogeneous responses to a subway line opening for cities with more economic activities. If the reduction in air pollution is more significant for these areas with more economic activities [36], it supports the claim that air quality improvement is due to the substitution of driving. In addition, they also test the mechanism by using a direct method, namely, estimating the relationship between the subway supply and on-road traffic based on traffic congestion data [41,42,43].

### 2.5. The Contributions of This Paper

However, the final effect on PM10 pollution after the new subway line openings is ambiguous. Even though the traffic diversion force reduces prolonged travel time on the road, and thus leads to the mitigation of PM10 pollution, as yet this decreasing force may be offset by the PM pollutants caused by the subway line supply, which likely increases PM10 pollution. Two recent papers reported that the PM concentrations in subway systems are higher than those in ambient air [14,44]. Among the various pollutants in subway systems worldwide, contamination by particulate matter (PM) is the most serious [41,45,46]. The mechanical abrasion of rail/wheel and brakes, and from the catenary, and the resuspension of material caused by air turbulence in the subway lines, are considered to be the main drivers for generating PM pollution [12,13]. Therefore, the change in PM10 pollution after the new subway openings in China is still unclear.

The main empirical methodology used in these emerging studies to estimate the air pollution effect after a new subway line opening is the RD method, which is widely applied in policy effect analysis. However, this approach is not able to examine the long-term equilibrium effect [11,47]. In addition, many previous studies have focused on investigating the change in air pollution after a subway line opening by using the DID method, but it is notable that the heterogeneity among cities would lead to the “differential deviation” of the DID method [16]. Furthermore, many studies have paid close attention to the mitigation of air pollution in either the short or medium term based on the case of individual cities, and not much of the literature focuses on estimating the mitigated air pollution effect after the new subway openings, in both the shorter and longer terms, at the city level, based on the case of Chinese cities.

Therefore, our paper tries to contribute to the emerging literature in three aspects. First of all, contrary to previous studies (such as [10,35]), it focuses on exploring both the short- and longer-term effects of the new subway openings on PM10 pollution at the city level in China, and then we suggest policy implications. Second, it uses the PSM-DID method, which differs from approaches introduced in the above-discussed studies (such as [11,36]), to investigate the effect of the subway line openings on PM10 pollution with a robustness check. Third, we also investigate the relationship between the subway line openings and PM10 pollution for cities with varying population sizes, and varying levels of economic development in China.

This paper organizes the following sections as follows. In Section 3, this paper presents the methods, data and sample. Section 4 discusses the results of estimation. Section 5 presents the discussion. Section 6 concludes.

## 3. Methods, Data and Sample

### 3.1. Empirical Strategy

In this section, we discuss our empirical methods. The main empirical framework employs the new subway line openings as the key explanatory variable, and uses the PSM-DID approach to address the sample selectivity bias and the individual heterogeneity. We present the difference-in-differences (DID) method and the propensity score matching (PSM) method in the following sections.

#### 3.1.1. Difference-In-Differences (DID) Method

DID method is a natural experiment approach, which is widely applied for estimating a policy’s effect. Due to the potential endogeneity and reverse causality issue of ordinary regression, the DID method can provide a unique opportunity to be able to avoid the occurrence of sample selectivity bias [11]. Compared to the ordinary regression (e.g., Ordinary Least Square method), a key step of the DID method is dividing the whole sample into two groups: the processing group, affected by the subway line openings, and the control group, without any subway line openings. The DID approach can control the difference between the processing group and the control group before and after the opening of a new subway line at the city level, as well as other systemic differences [16]. A basic DID framework is given by:(1)$$\ln(ap_{it})=\alpha_{0}+\alpha_{1}treated_{it}*T_{it}+\alpha_{2}treated_{it}+\alpha_{3}T_{it}+\alpha_{4}X_{it}+{}_{it}$$
where *ap_it_* is air pollutant concentration of city *i* at time *t*; *treated_it_* is a dummy variable that is equal to 1 if there is a new subway line opened in city *i*, and it is equal to 0 if there are not any subway openings in city *i*; *T_it_* is a time dummy variable that takes a value of 0 before a new subway line opening, and it takes a value of 1 after a new subway line opening; *X_it_* refers to weather variables, including average temperature, wind speed and relative humidity, and binary variables indicating rain, snow and storm; *ε**_it_* is the error term.

#### 3.1.2. Propensity Score Matching (PSM) Method

Even though the DID method has the advantage of overcoming self-selected sample bias, as yet there could still be bias if unobserved factors exist in the sample, i.e., different initial conditions between the processing group and the control group. For example, the Chinese cities that constructed subway lines were not random, but related to the level of economic development and the environmental conditions of those cities. Therefore, there is heterogeneity of cities between the processing group and the control group, which may result in “differential deviation” by the DID method. For reducing the sample selection bias and heterogeneity, and effectively matching the process group and the control group, the PSM method is introduced to handle with the selection bias caused by confounding factors [48]. In short, the combination of the DID with the PSM method can effectively estimate the causal effect of the new subway line openings on concerned outcomes—air quality in terms of PM10 concentration.

Decades ago, the first subway line in China opened in January 1971 in Beijing, and 35 cities (excluding Hong Kong, Macao and Taiwan) opened subway lines at the end of 2018. Due to the unavailability of air pollution data before December 2013, we choose 2014 as the time node. The city defines two aspects of the virtual variable, *treated_it_* = 1, or 0. *treated_it_* = 1 refers to the city i in the processing group, which includes cities with subway line openings during the data period; *treated_it_* = 0 refers to the control group, which includes cities without any subway line openings during the data period. The propensity score can turn a multidimensional variable into a one-dimensional variable, and thus obtain a value between 0 and 1, so that it can perfectly measure the difference of individual samples. *K* near neighbor matching and the logit method are usually used for computing the distance of samples and estimating the final propensity score.

### 3.2. Data and Sample

Table 1 presents the main variables of our analysis. Air quality in Chinese cities is measured by PM10 concentration, and we choose PM2.5 as a robustness pollutant because the relationship between the two is close (see Table 2; the coefficient between the two is high and the value is 0.886). All these indices are measured on a daily basis. In addition, the control variables contain daily weather variables and wind condition: average temperature (denoted by *qw*), average relative humidity (denoted by *xdsd*), binary variables indicating rain, snow and storm (denoted by *ifrain*), and average wind speed (denoted by *fs*). According to previous studies (e.g., [10]), average temperature and relative humidity tend to result in greater PM10 pollution, while average wind speed and rain would lead to a better PM10 environment.

The Ministry of Environmental Protection (MEP) has published hourly and daily air quality data every year since December 2013. The open air quality data include sulfur dioxide (SO_2_), nitrogen dioxide (NO_2_), carbon monoxide (CO), ozone (O_3_), coarse particles with a diameter of less than 10 µm (i.e., PM10), fine particles with a diameter of 2.5 µm or less (i.e., PM2.5) and Air Quality Index (AQI). In this paper, the key air pollutant is PM10, and PM2.5 is a robustness pollutant. These daily data of sample cities have been available from the MEP’s website since December 2013. The data of weather variables and wind conditions were collected from two key websites: https://www.wunderground.com/history/daily/ZSOF/date/2017-1-1 and http://data.sheshiyuanyi.com/WeatherData/.

The second data set records the opening dates of subway lines in the cities. We chose cities with a newly-opened subway line during the data period as our processing group of samples. The data of the opening dates of subways in these cities were collected from the China Metro website and the official websites of subway operating companies in the sample cities. During the data period from January 2014 to December 2017, the newly-built subway lines of 7 cities in China were opened.

We estimate the changes in air quality in terms of PM10 concentration before and after the new subway openings for Chinese cities. Therefore, the processing group contains 7 cities with open subway lines. The details of the processing group are shown in Table 3. Among the 215 cities for which data were collected, the 35 cities that had opened their subway lines between 2014 and 2017 were eliminated, and the remaining 181 cities were deemed samples in the control group. For better matching with the processing group, and to reduce the “differential deviation” originating from the whole sample, we first needed to obtain the data of the sample in the control group based on the control variables (including weather and wind variables indicated in the paper), using the Propensity Score Matching (PSM) method. However, among these 181 cities, we only obtained the full data of weather and wind variables in 20 cities, and thus we finally selected 20 cities from these 181 cities for the control group [49]. Accordingly, the 20 cities of the control group include Yancheng, Yangzhou, Huai’an, Taizhou, Jinhua, Jinchang, Quanzhou, Jilin, Zhuzhou, Kaifeng, Yichang, Weifang, Yibin, Baotou, Jiujiang, Xuchang, Nantong, Shangqiu, Rizhao and Zhu’madian.

## 4. Empirical Results

### 4.1. PSM Results

The first step of the analysis in this paper is to conduct the PSM method, in order to test the reliability of the matching results. The logit model was conducted with a set of control variables: average temperature, average relative humidity, binary variables indicating rain, snow and storm, and average wind speed. The *K* near neighbor matching method was used to match the sample cities. We then obtained the matching check for the distribution of the covariates between the processing group and the control group. The results are shown in Table 4.

Table 4 shows that the *p* value of each variable does not pass the significant test at the 5% significant level, which indicates that there is systematic difference between the processing group and the control group. Furthermore, according to the following Figure 1, the out-of-support untreated samples are small, and most of the observed values of the sample are on support, thus using the PSM method is considered to be suitable, and the result is reliable.

### 4.2. Regression Results

In this section, Table 5 presents the estimated impact of the new subway line openings using the OLS, DID and PSM-DID methods for a fixed short-term time window (i.e., 90 days). The key variable is the *treated**×**T*. We subsequently add weather variables and wind conditions as control variables. The results of column (1) show that the subway line openings result in high levels of PM10 concentration, using the OLS method. After controlling for the weather variables and wind condition, column (2) provides a similar result for the effect of the new subway openings on PM10 based on the same approach. Specifically, the value of R^2^ improves from 0.11 to 0.25, and the coefficient of *treated**×**T* is 0.15—a slight reduction compared with the result given by column (1). However, all of these control variables lead to the mitigation of PM10 pollution with the OLS method.

When we employ the PSM-DID method, the result from column (3) suggests that PM10 concentration experiences an increase (i.e., 18%) after the new subway’s openings. Compared with the result derived using the OLS method, column (3) displays a slight increase in the coefficient of *treated**×**T*. This may be due to an improvement in the matching between the processing group and the control group, and thus the reduction in sample selection bias.

Furthermore, the weather variables have intuitive signs: high levels of average temperature (denoted by *qw*) and relative humidity (denoted by *xdsd*) are associated with high levels of PM10 concentration, while rainfall/snow (denoted by *ifrain*) is associated with an improvement of air quality. However, high average wind speed (denoted by *fs*) leads to an increase in PM10 concentration.

According to meteorological theory, high temperature accelerates the chemical reactions that form ozone and secondary particulate matter, and thus leads to increasing PM pollution [50]. For example, in April 2016, people in India were exposed to the highest unhealthy levels of ozone (i.e., 150 µg/m^3^), which is 50% higher than the safe value given by the World Health Organization. The occurrence of this event comes after several days of high temperature, which accelerates the formation of ozone. Similarly, based on an outdoor air quality study, the American Lung Association found that more than half of the people in the country experienced a peak in both ozone and PM concentrations, and these were associated with increasing global temperatures [51]. Furthermore, higher humidity results in the adhesion of atmospheric PM on the water vapor that forms fog and suspends in the air, leading to the accumulation of air pollutants and aggravation of air pollution. However, once the air humidity rises to form the effective precipitation, then it has a scouring and scavenging effect on the air pollutant. Therefore, precipitation in the form of rainfall or snow can make PM10 pollution dissipate more quickly. However, a high average wind speed leads to concentrated PM10 pollutants, which is contrary to the result of Li et al. [10].

Column (4) shows that the results of the treatment effect PM2.5 pollution for a fixed time window (i.e., 90 days). Specifically, after controlling for those weather and wind variables, the treatment effect suggests a 34% increase in PM2.5 concentration based on the PSM-DID method. In addition, the results of the control variables confirm the robustness of the effect of subway line openings on PM10, because high average temperature, relative humidity and wind speed are associated with high PM2.5 concentrations, while rainfall/snow is associated with a reduction in PM2.5.

#### 4.2.1. Heterogeneity of Time Windows

Previous studies usually investigated the short- or medium-term mitigated air pollution effect of a new subway opening, based on the case of an individual city. We re-estimated both the short- and longer-term effects on PM10 using the case of Chinese cities. We chose 90 days, 350 days and 500 days before and after the new subway openings to estimate the treatment effect. The first processing group (90 days) contains Changsha, Ningbo and Wuxi. The second processing group (350 days) has Qingdao and Nanchang. The final processing group (500 days) includes Fuzhou and Nanning. After opening their first subway line, all these subway cities later opened their second subway line. The details of opening dates of the subway lines in these cities are shown in Table 3. The long time interval between the first and the second subway line provides us an opportunity to accurately estimate the impact of the subway line openings on PM10 pollution, by overcoming the interaction effects caused by the subway line openings that occurred at different times.

We estimated the treatment effects on both PM10 and PM2.5 pollutants. Table 6 reports regression results using different time windows (i.e., 90, 350 and 500 days) before and after the opening dates of subway lines. These significant estimates are statistically different across those different time windows [column (1)—90 days; column (2)—350 days; column (3)—500 days]. In the short term, the high levels of PM10 pollution after the new subway line openings are significant, and the coefficient is 0.21 [column (1)]. This result confirms the above result using the whole sample cities. Interestingly enough, this increasing PM10 effect seems to fade away if we increase the time windows. When we increase the time window to 350 days [column (2)], a 5% increase in PM10 pollution after the new subway line opening seems to be smaller than that of the 90-day window. When the window increases to 500 days, however, the new subway line openings exert a 9% reduction in PM10 pollution in the longer term, as shown in column (3). A possible explanation is that the subway-induced PM effect may offset the small substitution-of-driving effect in the short term, while the large substitution effect that increases with increasing time windows is overpowered at last, as the subways powered by electricity are relatively cleaner than gasoline vehicles, and thus the subway-induced PM is limited.

In addition, the control variables, including average temperature (denoted by *qw*), relative humidity (denoted by *xdsd*) and wind speed (denoted by *fs*), lead to high PM10 pollution, while the binary variable indicating rain, snow and storm (denoted by *ifrain*) tends to improve air quality. Similarly, all key variables in this paper exhibit the same effects on PM2.5 pollution within different time windows.

#### 4.2.2. Heterogeneity of Population Sizes

In China, population size is one of the key factors and standards that affects intra-city subway construction. More importantly, a recent study indicated that the level of traffic congestion may be different in cities with varying population sizes [52], and thus the effect of the new subway line openings on traffic congestion, and their final effects on PM10, may be different in these cities. Therefore, following Chang et al. [52], we divided the whole sample into three sub-samples: 1–2, 2–3 and 3–5 million permanent people within city. The three sub-samples were then divided by the level of permanent population sizes in order to estimate the heterogeneity of mitigating effects across these cities. The first processing group includes Nanchang; the second processing group contains Ningbo, Wuxi and Nanning; and the third processing group includes Changsha, Qingdao and Nanning.

Table 7 provides different results from the PSM-DID method, which account for the effect of new subway openings on both PM10 and PM2.5 concentrations for three sub-samples with varying population sizes. Specifically, for cities with 1–2 million permanent people, the subway openings exert an increased effect on PM10 [i.e., column (1), the coefficient is 0.24]. This increasing PM10 pollution may be due to the subway-induced PM effect taking over the small traffic diversion effect for cities with less than 2 million people in the short term. However, for cities with populations of over 2 million, the results suggest that the subway openings begin to improve air quality in terms of PM10 [i.e., columns (2) and (3)]. Specifically, for cities with 2–3 million people, column (2) shows a 2% reduction in PM10 pollution after the new subway openings; the results of column (3) show that air quality improvement increases to 3% for cities with over 3 million people—a slight increase in the improvement of air quality compared to cities with 2–3 million people. In short, the cities with more than 2 million people have high levels of air quality in terms of PM10 after the subway openings during the short term, probably due to an increase in the substitution of driving in these large cities with larger numbers of people.

Similarly, for a fixed time window, the results in columns (4)–(6) show the same effect on PM2.5 concentration. Specifically, high PM2.5 pollution levels after the subway openings for cities with 1–2 million people, and an improvement of air quality for cities with over 2 million people.

The control variables, including average temperature and relative humidity, result in high PM10 pollution, while binary variables indicating rain, snow and storm, and average wind speed, improve air quality for cities with 1–2 million people. However, for those cities with over 2 million people, average wind speed tends to result in high PM10 and PM2.5 pollutants.

#### 4.2.3. Heterogeneity of the Level of Economic Development

In addition to population sizes, the level of economic development of the city can affect a city’s subway construction. Furthermore, cities with high levels of economic development may have more social and economic interactions, and more traffic congestion. Therefore, the effect of the new subway line opening on traffic congestion, and its final effect on PM10 pollution, may be different for cities with varying levels of economic development. Therefore, we divided the processing cities into two groups: cities with GDP of CNY 3000–8000 million, and cities with GDP of CNY 8000–10,000 million. The first processing group includes Nanchang, Nanning, Changsha, Ningbo and Fuzhou, and the second processing group contains Wuxi and Qingdao.

Table 8 reports the results of the treatment effect with regards to both PM10 and PM2.5 using the PSM-DID method. Column (1) in Table 8 suggests that the opening of subways improves air quality for cities with GDP of CNY 3000–8000 million. PM10 pollution reduces by about 10% after the subway line openings. Column (2) also shows that the new subway openings do significantly improve air quality for cities with GDP of CNY 8000–10,000 million, and the treatment effect increases to about 22%. The control variables exert the same results, as shown in the above section, namely average temperature, relative humidity and wind speed variables result in high PM10 pollution, and binary variables indicating rain, snow and storm tend to improve air quality.

The treatment effect on PM2.5 pollution also shows the same significant and negative relationship between the two factors for varying levels of GDP. In those cities with high levels of GDP, the mitigation of PM pollution after the new subway openings confirms the above discussion that the traffic diversion effect may take over the subway PM effect.

### 4.3. Robustness Check

In addition to PM2.5 as a robust pollutant, Table 9 in this section reports our examining of the evidence for the robustness of our results, using two alternative specifications—the OLS and DID methods.

Columns (1) and (3) show the results for both PM2.5 and PM10 pollutants using the OLS method for a fixed time window (i.e., 90-day). Furthermore, due to the processing group used in estimating the effect of the new subway line openings on PM10 concentration including cities which opened a second new subway line a short time after, we excluded these cities and obtained a new processing group (i.e., Nanchang, Fuzhou and Nanning), in order to avoid interaction effects caused by subway lines that opened at different times [11]. Using these new sample cities, we estimated the effects of new subway openings on PM10 and PM2.5. The results show that the new subway openings lead to high PM10 and PM2.5 pollution, which results are consistent with those for PM10 and PM2.5 based on the PSM-DID method.

Columns (2) and (4) show the results for both PM2.5 and PM10 pollutants using the DID method for a fixed time window (i.e., 500-day). The findings of the DID method, however, indicate that the subway openings result in high PM2.5 and PM10 pollutants. The difference between the results of the DID method and those of the PSM-DID method can be explained by the issue of “differential deviation” originating from the heterogeneity among cities. Using the PSM method, we obtained a new sample in the control group based on the initial conditions including weather conditions and wind speed, leading to the improvement of the estimating of the model given better matching between the processing group and the control group.

### 4.4. Mechanism Analysis

We suppose that the subway line openings can increase (reduce) PM10 pollution if the subway openings have a little (large) effect on the substitution of driving, and if the subway-induced PM pollution is limited. The latter is obvious, since the subways powered by electricity are relatively cleaner than gasoline vehicles.

We tested the substitution-of-driving effect by estimating both the short- and long-term effects of the subway openings on CO, which is a major pollutant emitted by motor vehicles. In 2016, the quantity of vehicle emissions in China was 44,725 million tons, of which 34.19 million tons were CO and 0.53 million tons were PM [53]. Therefore, we estimated the effect of the new subway openings on CO in order to test the alternatives to subway transit in the form of on-road transportation. The results in Table 10 show that the new subway line openings result in the mitigation of CO in both the shorter and longer term. Specifically, CO experienced nearly 10% (for the 90-day time window), 26% (for the 350-day window) and 62% (for the 500-day window) reductions, respectively. The increasing reduction in CO with increasing time windows prove that the subway line openings substitutes on-road transportation to some extent, namely, the little substitution-of-driving effect in the short term, and a relatively large effect in the longer term.

In addition, for fixed time windows (i.e., short-term), columns (4)–(6) show an increasing mitigation of CO pollution (i.e., 28% for cities with 1–2 million people, 38% for cities with 2–3 million, and 42% for cities with 3–5 million) after the subway openings. Thus indicates an increasing substitution-of-driving effect for cities with increasing population sizes. Furthermore, columns (7) and (8) find 59% and 58% reductions in CO pollution for cities with GDP of CNY 3000–8000 million and 8000–10,000 million after the subway openings, respectively. This indicates that there is a large substitution-of-driving effect for cities with different levels of economic development in the short term.

## 5. Discussion

The current research takes a first step towards a better understanding of the relationship between the new subway line openings and air quality, in terms of PM10 concentration, for a sample of Chinese cities, by making use of the PSM-DID method. Due to the issues of the ordinary regression method, the PSM-DID method provides a unique opportunity to avoid the occurrence of sample selectivity bias and individual heterogeneity, and the credibility of the results in this paper is improved by a robustness check. The analysis indicates that air quality in terms of PM10 pollution in these cities responds to the subway line openings in a way that is different than what was expected from earlier studies, based on the cases of individual cities.

The results show that the new subway line openings do not seem to be a good strategy to reduce both PM10 and PM2.5 concentrations for Chinese cities in the short term. Chinese cities with more subway line openings tend to present high PM10 and PM2.5 concentrations, contrary to previous studies based on individual case, such as Li et al. [10] and Zheng et al. [11], in which the results suggest that a new subway opening improves air quality. The difference can be explained by the fact that the small traffic diversion effect may be offset by the subway PM effect in the short term, which thus results in high PM pollution in cities. On the one hand, many published studies about subway systems have indicated an average PM10 concentration caused by subway lines that exceeds 50 µg/m^3^, and even 300 µg/m^3^ in some cases [54,55], which can be higher than those above ground [12]. The high levels of indoor air pollution can also create extremely high levels of outside air pollution.

On the other hand, the subway line openings seem to have a small effect on the substitution of driving during the short term, in the above mechanism analysis. In addition, there is other evidence supporting the small substitution effect during the short term. According to the commuting data obtained from the AutoNavi Traffic Big-data Team of AutoNavi Software Co., taking a subway line was faster than driving for commuting people in Changsha in 48% of cases, while it was quicker to go by car than to take a subway in Wuxi and Ningbo in the 80% of cases [56] in the first quarter of 2016 after the subway openings. In general, people are concerned about whether taking the subway is faster than driving. In this regard, the subway openings have little effect on the substitution-of-driving effect in these cities during the short term.

The effect of the new subway line openings on PM10 pollution for sub-sample cities in different time windows is also interesting. In previous studies, Li et al. [10] showed that the new subway opening improves air quality in the short term (i.e., the 60-day time window), while this mitigating effect fades away when they increase the window to 110 days and longer. However, using the data of Chinese cities, this paper suggests that the increasing effects on PM pollution are not statistically different between the 90-day and the 350-day windows (namely, the short term). This result is consistent with the above conclusion for the whole sample cities. However, when we increase the window to 500 days, the positive relationship between the subway line openings and PM10 pollution during the short term seems to turn negative. The statistically different results between the short and longer terms can be explained by the fact that the new subway line openings may not have the greatest effect on traffic congestion and PM pollution in the short term, as most commuters may drive as they did before they relocated into metro neighborhoods [57], and thus the subway PM effect is dominated.

Due to limited data availability, we tested the substitution of driving by presenting the traffic mode split data of Beijing from 2011 to 2016 (as shown in Figure 1). In 2014, Beijing opened four subway lines, with 6, 7, 14 and 15 lines. Figure 2 reports that the proportion of people travelling by subway, transit bus, car, taxi and bicycle increased after the subway openings in 2014. Although the rise in the proportion of people that take the subway has been relatively stable after the four subway line openings, there has also been a slight increase in the proportion of driving people. Therefore, the subway openings appear to have no significant reduction effect on the shift from private car usage to subway usage. However, in the longer term, the further reduction of traffic congestion results from less driving of commuters (namely, the large substitution-of-driving effect) who relocated into metro areas, which could lead to a mitigation of the initial increase in PM10 pollution, because commuting people can adapt and make longer-run decisions, such as relocating residential or employment locations in order to better utilize subway system [47].

Another surprising result corresponds to the effect on PM10 pollution after the new subway line openings for cities with varying population sizes. The different results between cities with 1–2 million people (i.e., positive) and cities with over 2 million people (i.e., negative) may be due to the following reason. The PM effect may take over the small traffic diversion effect for cities with 1–2 million people, and thus lead to an increase in PM10 pollution after the new subway openings. However, for cities with over 2 million people, the increasing mitigation of traffic congestion may be dominated, which is likely to reduce PM10 pollution. Furthermore, the fact that the mitigation of PM10 pollution tends to increase with population sizes may be due to the fact that substitution of driving, stemming from the new subway line openings, is more significant for these cities with lager populations.

The results regarding whether wind speed can improve air quality in terms of PM are controversial. The prevailing view in the field of air pollution research is that wind speed tends to improve air quality, as indicated in Li et al. [10]. However, other studies (i.e., [50]) find that there is no relationship between average wind speed and air quality in Beijing. The result in this paper indicates that average wind speed seems to lead to high PM10 pollution. The difference can be explained by the incomparability of high-rise buildings in high density cities [58]. For cities with larger population sizes, building heights in these high-density cities are greater, which may explain why high wind speed does not lead to a good PM environment, as this may be influenced by an increase in high-rise buildings. A recent paper indicated that the corner wind zones caused by high rise buildings in high-density urban areas often sharply increase wind speed and generate strong winds that bring up dust from the ground and form new dust pollution [58].

## 6. Conclusions

The issue of severe traffic congestion and air pollution in Chinese cities has attracted the Chinese government’s attention. In addition to demand-side measures, such as the car restricting policy, large investment in public transport infrastructure, such as subway systems construction, is one of the main strategies for improving air quality and maintaining human health [23]. China is now constructing thousands of kilometers of subway systems in major cities [59].

While previous research has examined the mitigated congestion effect of public transport infrastructure and the impact of a subway line opening on air quality, based on the case of individual cities, there is limited evidence regarding the short- and longer-term relationships between the new subway line openings and PM10 pollution at the city level in China. Furthermore, the investigation into the relationship between a new subway line opening and air quality usually uses the RD and DID methods, rather than the PSM-DID method. To our knowledge, the ordinary regression method would produce a sample selectivity bias, which may lead to the effect of the difference in the predictions, while the PSM-DID method used in this paper may avoid the occurrence of sample selectivity bias and heterogeneity, and obtain more robust results.

Using daily PM10 concentration data and the new subway openings of 7 Chinese cities from January 2014 to December 2017, based on the PSM-DID method, we find that the openings of subway lines seems to lead to high PM10 concentrations in the short term, using the whole sample cities. Furthermore, the result of the treatment effect on PM10 pollution for sub-sample cities with varying time windows confirms the above result, namely, an increase in PM10 pollution after the subway openings in the short term. However, for the longer time windows, the opening of subway lines tends to improve air quality in terms of PM10. Unlike previous studies, which indicate that the new subway line opening improves air quality in the short or medium term, based on the case of individual cities (e.g., Li et al. [10] and Zheng et al. [11]), our findings stress that the new subway line openings result in high levels of PM10 concentration in the short term, while the longer-term effect is improved air quality.

In addition, we examine the relationship between the new subway line openings and PM10 pollution for sub-samples cities with varying population sizes. We find that the new subway line openings result in increasing PM10 pollution for cities with 1–2 million people population sizes. However, for cities with over 2 million people, it leads to the reduction of PM10 pollution. We also estimate the treatment effect on PM10 concentration for cities with varying levels of economic development. We show that the reduction of PM10 pollution is significant for cities with GDP of over CNY 3000 million.

Furthermore, our findings for the heterogeneous effects show that the reduction in CO pollution is significantly lower after the new subway openings during the short term, but is relatively larger with increasing time windows. Overall, these findings indicate that the new subway line openings have led large numbers of people to switch from driving to subway, thus resulting in the improvement in air quality in terms of PM10 in the longer term. However, it should be noted that this improvement may be small due to the subway-induced PM pollution effect. The robustness checks are also conducted based on PM2.5 concentration; the relationship between new subway openings and air quality is still robust.

There are also two limitations in our study. First, we recognize that our framework has not incorporated a number of variables that may affect air pollution in urban areas. For example, a number of local factors may affect air pollution differently across cities, which may include demand-side strategies, such as the driving restriction policy, the car license plate restriction policy, and many other factors. Second, due to limited data availability, we investigate the substitution-of-driving effect by estimating the impact of the subway line openings on the major pollutant emitted by motor vehicles—CO. However, with the application of big data technology in the transport area, more commuting data can be obtained in the future, and thus the influence mechanism of the subway line openings on PM pollution needs further study.

Notwithstanding the limitations of this paper, we try to shed some new light on the effect of new subway openings on PM10 concentration for a sample of Chinese cities, which is important for supporting evidence-based public transport infrastructure construction. Our result suggests that the current subway construction in China contributes to improving air quality in terms of PM10 and PM2.5 concentrations in the longer term. Moreover, public transport infrastructure investment should be directed towards cities with over 2 million people in order to reduce PM pollution. However, these reductions in PM may be small, due to the subway-induced PM pollution.

## Figures and Tables

**Figure 1 ijerph-17-04638-f001:**
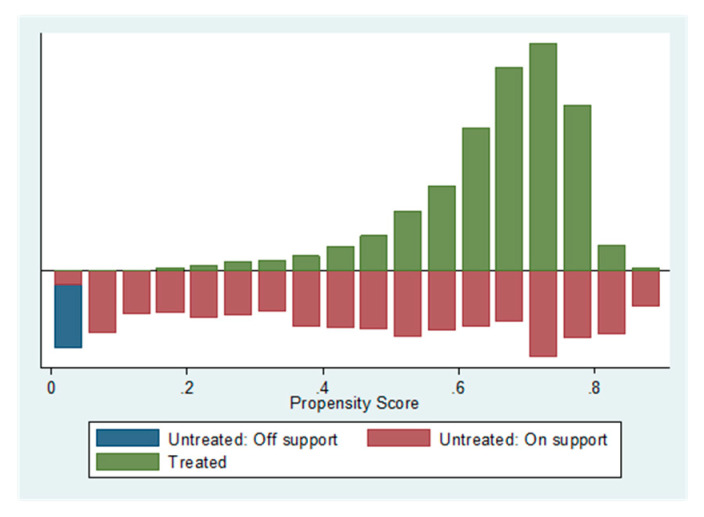
The PSM validity test 2: PM10.

**Figure 2 ijerph-17-04638-f002:**
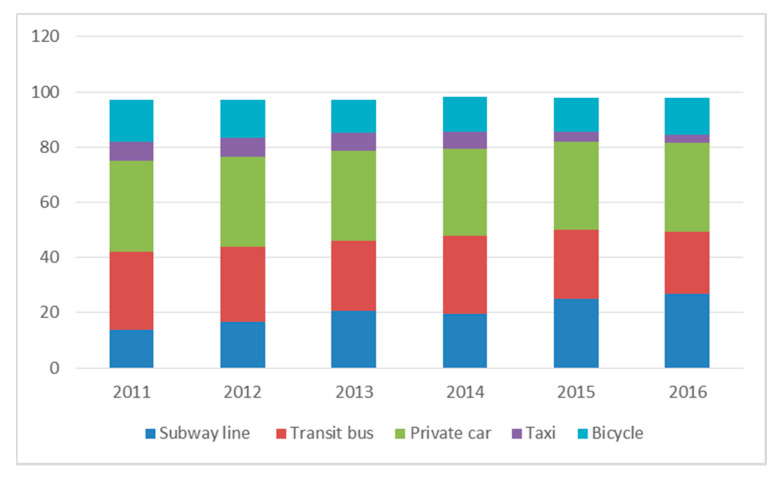
Traffic mode split distribution in Beijing from 2011 to 2016.

**Table 1 ijerph-17-04638-t001:** Variable definitions and summary statistics.

Variable.	Obs	Mean	Std. Dev.	Min	Max	Unit	Definition
ln(PM2.5)	12,090	3.711	0.683	1.386	6.084	mg/m^3^	Daily PM2.5 concentration
ln(PM10)	12,034	4.302	0.611	0	7.489	mg/m^3^	Daily PM10 concentration
ln(CO)	12,068	0.872	1.597	−1.609	4.771	mg/m^3^	Daily CO concentration
*treated*	12,170	0.551	0.497	0	1	-	-
*T*	12,170	0.501	0.500	0	1	-	-
*ifrain*	12,133	0.378	0.485	0	1	-	Rain/snow/storm dummy: 1 if there was rain/snow/storm, 0 otherwise
ln(*qw*)	12,138	6.800	0.706	0	7.569	°C	Mean temperature
ln(*xdsd*)	12,138	6.491	0.684	0	7.122	%	Relative humidity
ln(*fs*)	12,138	2.224	1.311	0	18.624	m/s	Mean wind speed

**Table 2 ijerph-17-04638-t002:** The correlation between variables.

Variable	ln(PM2.5)	ln(PM10)	ln(CO)	*treated*	*T*	*ifrain*	ln(*qw*)	ln(*xdsd*)	ln(*fs*)
ln(PM2.5)	1.000								
ln(PM10)	0.886	1.000							
ln(CO)	0.308	0.288	1.000						
*treated*	−0.273	−0.260	−0.705	1.000					
*T*	−0.209	−0.157	0.009	−0.120	1.000				
*ifrain*	−0.260	−0.346	−0.006	−0.022	0.087	1.000			
ln(*qw*)	−0.142	−0.121	−0.146	0.092	0.187	0.048	1.000		
ln(*xdsd*)	−0.157	−0.298	−0.212	0.171	0.139	0.378	0.180	1.000	
ln(*fs*)	−0.129	−0.077	−0.100	0.092	−0.019	−0.067	−0.089	−0.235	1.000

**Table 3 ijerph-17-04638-t003:** The characteristics of the processing group.

City	Province	Annual Populations (Ten Thousand People)	Gross Domestic Production (a Hundred Million Yuan)	Opening Date of the Opening Date of the First Subway Line	Opening Date of the Second Subway Line
Changsha	Hunan	304	7825	29 April 2014	21 March 2016
Ningbo	Zhejiang	230	7610	30 May 2014	26 September 2015
Wuxi	Jiangsu	246	8205	1 July 2014	28 December 2014
Qingdao	Shandong	480	9300	16 December 2015	10 December 2017
Nanchang	Jiangxi	178	4000	26 December 2015	30 June 2019
Fuzhou	Fujian	203	6197	18 June 2016	26 April 2019
Nanning	Guangxi	370	3703	28 July 2016	6 June 2019

Data source: the data of annual populations and gross domestic production (GDP) from 2014 to 2016 was from the China City Statistical Yearbook.

**Table 4 ijerph-17-04638-t004:** The PSM validity test 1: PM10.

Variable	Unmatched/Matched	Mean		%Bias	%Reduct |Bias|	t Test	
		Treated	Control			t	*p* > |t|
ifrain	U	0.448	0.399	9.9		4.74	0.000
	M	0.445	0.447	−0.3	96.9	−0.14	0.892
ln(*fs*)	U	0.604	0.564	8.3		4.01	0.138
	M	0.567	0.644	−16.0	−93.2	−7.36	0.000
ln(*xdsd*)	U	4.306	4.281	11.6		5.59	0.000
	M	4.302	4.302	0.0	99.9	0.00	0.997
ln(*qw*)	U	3.062	2.556	83.4		40.95	0.000
	M	3.006	2.932	12.1	85.5	8.07	0.687

**Table 5 ijerph-17-04638-t005:** OLS, DID and PSM-DID estimates with a fixed time window (i.e., 90 days).

	ln(PM10)			ln(PM2.5)
	(1) OLS	(2) OLS	(3) PSM-DID	(4) PSM-DID
*treated×T*	0.23 (10.42) ***	0.15 (7.17) ***	0.175 (8.03) ***	0.344 (14.79) ***
ln(*qw*)		−0.03 (−3.06) **	0.59 (15.72) ***	0.59 (15.66) ***
ln(*xdsd*)		−0.14 (−14.71) ***	0.87 (17.40) ***	0.85 (17.08) ***
ln(*fs*)		−0.18 (−17.40) ***	0.56 (9.80) ***	0.54 (9.59) ***
*ifrain*		−0.36 (−32.84) ***	−0.23 (−3.54) ***	−0.24 (−3.58) ***
constant	4.69 (351.16) ***	5.94 (77.50) ***	−9.31 (−23.45) ***	−9.14 (−23.21) ***
Time window (days)		τ ± 90	τ ± 90	τ ± 90
N	12,034	11,990	5964	5997
R^2^	0.11	0.25	0.11	0.16

Significance: * *p* < 0.1, ** *p* < 0.05, and *** *p* < 0.01.

**Table 6 ijerph-17-04638-t006:** PSM-DID estimates with varying time windows.

	ln(PM10)			ln(PM2.5)		
	(1) PSM-DID	(2) PSM-DID	(3) PSM-DID	(4) PSM-DID	(5) PSM-DID	(6) PSM-DID
*treated×T*	0.21 (6.97) ***	0.05 (3.34) **	−0.09 (6.25) ***	0.15 (5.01) ***	0.08 (4.06) **	−0.05 (3.09) ***
ln(*qw*)	0.41 (5.80) ***	0.15 (3.90) ***	1.12 (18.71) ***	0.41 (5.74) ***	0.16 (3.96) ***	1.13 (18.76) ***
ln(*xdsd*)	3.63 (12.70) ***	0.14 (3.24) ***	2.39 (14.63) ***	3.56 (12.55) ***	0.14 (3.16) ***	2.38 (14.58) ***
ln(*fs*)	0.88 (7.95) ***	0.63 (10.87) ***	0.16 (3.15) **	0.88 (7.91) ***	0.63 (10.84) ***	0.16 (3.07) ***
*ifrain*	−1.28 (−9.48) ***	−0.61 (−8.93) ***	−0.14 (−2.47) **	−1.28 (−9.55) ***	−0.62 (−8.98) ***	−0.14 (−2.50) ***
constant	−19.97 (−15.73) ***	−3.68 (−12.40) ***	−15.48 (−21.42) ***	−19.69 (−15.60) ***	−3.67 (−12.36) ***	−15.45 (−21.39) ***
time window (days)	τ ± 90	τ ± 350	τ ± 500	τ ± 90	τ ± 350	τ ± 500
N	3384	12,902	15,411	3428	12,959	15,468

Significance: * *p* < 0.1, ** *p* < 0.05, and *** *p* < 0.01.

**Table 7 ijerph-17-04638-t007:** PSM-DID estimates with varying population sizes.

	ln(PM10)			ln(PM2.5)		
	(1) PSM-DID	(2) PSM-DID	(3) PSM-DID	(4) PSM-DID	(5) PSM-DID	(6) PSM-DID
*treated×T*	0.24 (15.07) ***	−0.02 (1.78) *	−0.03 (2.04) **	0.15 (8.05) ***	−0.02 (1.53) *	−0.01 (0.46) *
ln(*qw*)	0.45 (6.37) ***	0.30 (6.49) ***	0.41 (9.96) ***	0.46 (6.77) ***	0.30 (6.49) ***	0.41 (10.01) ***
ln(*xdsd*)	3.20 (12.11) ***	1.67 (10.91) ***	0.39 (8.59) ***	3.19 (12.07) ***	1.66 (10.85) ***	0.38 (8.52) ***
ln(*fs*)	−0.29 (−4.10) ***	0.65 (11.30) ***	0.43 (−18.75) ***	−0.29 (−4.16) ***	0.65 (11.26) ***	0.43 (8.37) ***
*ifrain*	−0.58 (−6.70) ***	−0.17 (−2.57) **	−0.59 (−9.83) ***	−0.58 (−6.70) ***	−0.17 (−2.58) **	−0.59 (−9.87) ***
constant	−17.57 (−15.27) ***	−12.04 (−16.93) ***	−5.73 (−18.75) ***	−17.52 (−15.24) ***	−12.00 (−16.88) ***	−5.71 (−18.71) ***
Time window (days)	τ ± 350	τ ± 90	τ ± 350	τ ± 350	τ ± 90	τ ± 350
population(million)	1–2	2–3	3–5	1–2	2–3	3–5
N	14,331	16,254	15,290	14,388	16,311	15,347

Significance: * *p* < 0.1, ** *p* < 0.05, and *** *p* < 0.01.

**Table 8 ijerph-17-04638-t008:** PSM-DID estimates with different levels of GDP.

	ln(PM10)		ln(PM2.5)	
	(1) PSM-DID	(2) PSM-DID	(3) PSM-DID	(4) PSM-DID
*treated×T*	−0.102 (7.32) ***	−0.216 (15.66) ***	−0.100 (6.60) ***	−0.178 (11.80) ***
ln(*qw*)	0.310 (7.53) ***	−0.388 (−4.86) ***	0.310 (7.54) ***	−0.386 (−4.84) ***
ln(*xdsd*)	2.460 (16.62) ***	1.247 (3.13) ***	2.448 (16.56) ***	1.230 (3.09) ***
ln(*fs*)	0.154 (3.52) ***	0.010 (0.11)	0.150 (3.44) ***	0.008 (0.09)
*ifrain*	−0.156 (−2.90) ***	−1.478 (−6.04) ***	−0.156 (−2.91) ***	−1.478 (−6.04) ***
constant	−14.503 (−21.11) ***	−8.357 (−4.62) ***	−14.456 (−21.06) ***	−8.284 (−4.59) ***
Time window (days)	τ ± 90	τ ± 90	τ ± 90	τ ± 90
GDP (a hundred million)	3000–8000	8000–10,000	3000–8000	8000–10,000
N	14,622	11,679	14,679	11,754

Significance: * *p* < 0.1, ** *p* < 0.05, and *** *p* < 0.01.

**Table 9 ijerph-17-04638-t009:** Robustness check: estimation for PM2.5 and PM10 pollutants.

	ln(PM2.5)		ln(PM10)	
	(1) OLS	(2) DID	(3) OLS	(4) DID
*treated×T*	0.17 (6.29) ***	0.10 (4.77) ***	0.16 (6.31) ***	0.04 (2.03) **
ln(*qw*)	−0.18 (−13.18) ***		−0.03 (−2.23) **	
ln(*xdsd*)	−0.002 (−0.07)		−0.41 (−12.42) ***	
ln(*fs*)	−0.26 (−19.75) ***		−0.22 (−17.97) ***	
*ifrain*	−0.32 (−24.48) ***		−0.34 (−28.04) ***	
constant	4.87 (34.31) ***		6.73 (48.69) ***	
time window(days)	τ ± 90	τ ± 500	τ ± 90	τ ± 500
population(million)				
N	9337	26,811	9281	26,755
R^2^	0.29	0.19	0.30	0.20

Significance: * *p* < 0.1, ** *p* < 0.05, and *** *p* < 0.01.

**Table 10 ijerph-17-04638-t010:** The impact of the subway openings on CO.

	ln(CO)							
	(1) PSM-DID	(2) PSM-DID	(3) PSM-DID	(4) PSM-DID	(5) PSM-DID	(6) PSM-DID	(7) PSM-DID	(8) PSM-DID
*treated×T*	−0.10 (2.02) **	−0.26 (10.41) ***	−0.62 (20.63) ***	−0.28 (8.81) ***	−0.38 (13.33) ***	−0.42 (14.63) ***	−0.593 (20.98) ***	−0.578 (19.19) ***
ln(*qw*)	2.25 (28.01) ***	0.15 (3.82) ***	1.12 (18.72) ***	0.45 (6.74) ***	0.30 (6.46) ***	0.41 (9.97) ***	0.308 (7.49) ***	−0.388 (−4.86) ***
ln(*xdsd*)	3.76 (15.26) ***	0.18 (3.54) ***	2.37 (14.54) ***	3.18 (12.04) ***	1.65 (6.46) ***	0.38 (8.47) ***	2.438 (16.48) ***	1.219 (3.06) ***
ln(*fs*)	0.19 (2.51) **	0.64 (10.96) ***	0.16 (3.09) **	−0.29 (−4.14) ***	0.65 (11.27) ***	0.43 (8.39) ***	0.151 (3.46) ***	0.009 (0.10)
*ifrain*	−0.36 (−4.02) ***	−0.63 (−9.18) ***	−0.14 (−2.54) **	−0.58 (−6.73) ***	−0.18 (−2.61) ***	−0.59 (−9.91) ***	−0.158 (−2.94) ***	−1.480 (−6.05) ***
constant	−21.66 (−19.97) ***	−3.87 (−11.81) ***	−15.41 (−21.33) ***	−17.49 (−15.20) ***	−11.97 (−16.83) ***	−5.69 (−18.64) ***	−14.395 (−20.96) ***	−8.234 (−4.56) ***
time window(days)	τ ± 90	τ ± 350	τ ± 500	τ ± 350	τ ± 90	τ ± 90	τ ± 90	τ ± 90
population				1–2	2–3	3–5		
GDP							3000–8000	8000–10,000
N	4370	28,335	15,422	14,342	16,257	15,301	14,625	11,708
R^2^	0.56	0.50	0.52	0.49	0.53	0.51	0.51	0.50

Significance: * *p* < 0.1, ** *p* < 0.05, and *** *p* < 0.01.

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
