# Peer review of "Can the New Subway Line Openings Mitigate PM10 Concentration? Evidence from Chinese Cities Based on the PSM-DID Method"

_ijerph, 2020, doi:10.3390/ijerph17134638_

Round 1
Reviewer 1 Report
I appreciate the revisions by authors and believe you have made a good effort to address my comments. There will always be areas for improvement, but a disclosure of limitations of your study will be helful.
As far as the methodology used, the paper is fine. The lack of data may suggest that further research on the topic has to be done, and you could say so in the conclusions.
Author Response
Dear Reviewer 1:
We are very thankful and excited to have been given the opportunity to revise our manuscript, titled “Can the new subway line openings mitigate PM10 concentration?-Evidence from Chinese cities based on the PSM-DID method” and ID is “ijerph-623826” for International Journal of Environmental Research and Public Health. We carefully considered the comments advised by your kind self. Those comments are all valuable and very helpful for revising and improving our paper, as well as the important guiding significance to our researches. We have studied comments carefully and have made correction which we hope meet with approval. For your kind consideration, revised parts have been revised by using the “Track Changes” in the revised version. In addition, we have made careful examination in the original manuscript and we also have consulted native English speakers for paper revision. According to their suggestion, we have already corrected the grammar mistakes appeared in original version and used more clear term in the revised version. The main corrections in the paper and responds to your valuable comments are as following:
Responds to your comments:
Point 1: I appreciate the revisions by authors and believe you have made a good effort to address my comments. There will always be areas for improvement, but a disclosure of limitations of your study will be helpful. As far as the methodology used, the paper is fine. The lack of data may suggest that further research on the topic has to be done, and you could say so in the conclusions.
Response 1: Thank you so much for your kind suggestion. As suggested by the reviewer, lines 599-603 in the revised version have been stressed the limitation of this paper, namely the influence mechanism of the subway line openings on PM pollution needs further study (See lines 599-603).
Once again, thank you very much for your comments and suggestions.
We tried our best to improve the manuscript and made some changes in the manuscript according to the reviewers’ comments. These changes will not influence the content and framework of the paper.
We appreciate for Editors/Reviewers’ warm work earnestly, and hope that the correction will meet with approval.
Once again, thank you very much for your comments and suggestions.
We look forward to hearing from you in due time regarding our submission and to respond to any further questions and comments you may have.
Yours Sincerely,
Corresponding Author

Reviewer 2 Report
The article is well presented and discussed. It deals with the relationship between public underground transportation and Particulate Matter levels, using the Propensity Score Matching – Difference in Differences method. The background of the statistical method is discussed, as well as the results.
However, some issues are related to a clear scarce background knowledge of the mechanisms of production, transport and removal of aerosol. The main one concerns the supposed PM effect of public transportation: observations are reported in the following.
The language is not always clear; attentive revision of English is solicited. In particular, the use of terms as ‘positive’, ’negative’, ‘poor’, associated to PM or air pollution, is not clear. Please, use instead ‘high’ and ‘low’ adjectives associated to ‘air quality’ or ‘PM load’ which refer to a more quantitative category and are easier to understand.
Some observations:
line 14: ‘scare’ => maybe the authors meant ‘scarce’
30: ‘have’. Phrase is not clear, please clarify
50: The cited paper actually founds no evidence in PM decrease.
52: ‘high’ => higher.
54: The statements above (referring to refs 11, 12, 13) do not mean that the subway is a source of particulate: though the PM concentration is higher indoor, this does not contribute to PM levels outdoor. This is an important point since it is cited many times in the following (e.g.: 353; 379; 484-485; 419; 509; 527). The statement is strong: if the authors trust it, they should provide more clues.
59: why weather variables are unavailable? The authors use them in the following.
83: ‘poor air pollution’ is not clear, please clarify
89: again, ‘negative’ is not clear in this context
95: ‘healthy’
119,131 ‘percent’ => %
150: ‘it is support that…’ is not clear, please clarify
200: ‘equals to 1’ => ‘that is equal to’ or ‘that equals 1’
348: The statement does not make sense, please rephrase it.
474: ‘provide’ => provides
524: ‘another the surprising’ => another surprising
Author Response
Dear Reviewer 2:
We are very thankful and excited to have been given the opportunity to revise our manuscript, titled “Can the new subway line openings mitigate PM10 concentration?-Evidence from Chinese cities based on the PSM-DID method” and ID is “ijerph-623826” for International Journal of Environmental Research and Public Health. We carefully considered the comments advised by your kind self. Those comments are all valuable and very helpful for revising and improving our paper, as well as the important guiding significance to our researches. We have studied comments carefully and have made correction which we hope meet with approval. For your kind consideration, revised parts have been revised by using the “Track Changes” in the revised version. The main corrections in the paper and responds to your valuable comments are as following:
Responds to your comments:
Point: The language is not always clear; attentive revision of English is solicited. In particular, the use of terms as ‘positive’, ‘negative’, ‘poor’, associated to PM or air pollution, is not clear. Please, use instead ‘high’ and ‘low’ adjectives associated to ‘air quality’ or ‘PM load’ which refer to a more quantitative category and are easier to understand.
Response: According to your suggestion, we have made careful examination and we also have consulted native English speakers for paper revision before the submission this time. We have already corrected the grammar mistakes appeared in original version and used more clear term in the revised version (See lines 18-23, 51, 86, 91-93, 302, 303-304, 310-311, 332-333, 364-365, 384, 386-387, 390, 391-392, 396-397, 398-400, 401, 414, 416-417, 419-420, 438, 442, 460, 488, 507, 573-579).
Point 1: ‘scare’ => maybe the authors meant ‘scarce’ (as shown on line 14)
Response 1: We are very sorry for our negligence of the right spelling in the paper. As suggested by the reviewer, line 14 has been corrected in revised version (See line 14).
Point 2: ‘have’. Phrase is not clear, please clarify (as shown on line 30 )
Response 2: As suggested by the reviewer, line 30 in revised version has been clarified (See lines 31-33).
Point 3: The cited paper actually founds no evidence in PM decrease. (as shown on line 50)
Response 3: Thank you so much for your kind suggestion. We try to make a clarification here. Li et al. (2019) (namely, reference [10]) investigate the impact of subway system on air quality by leveraging fine-scale air quality data from 2008 to 2016 and find that the subway openings improves air quality based on DID method. To our best knowledge, the air quality variable in Li et al. (2019) is measured by air quality index (AQI), which is composed of SO2, NO2, PM10, PM2.5 and O3. Due to the subway line openings improve air quality in terms of AQI, we believe that it can mitigate traffic congestion and thus improve air quality in terms of PM10.
Point 4: ‘high’ => higher. (as shown on line 52)
Response 4: We are so sorry for our negligence of the correct spelling in the paper again. After careful examination, we have revised this grammar mistake in the revised paper (See line 54).
Point 5: The statements above (referring to refs 11, 12, 13) do not mean that the subway is a source of particulate: though the PM concentration is higher indoor, this does not contribute to PM levels outdoor. This is an important point since it is cited many times in the following (e.g.: 353; 379; 484-485; 419; 509; 527). The statement is strong: if the authors trust it, they should provide more clues. (as shown on line 54)
Response 5: Thank you so much for your kind suggestion. We try to make a clarification here. Some researchers have reported that the concentration of particles in the subway were higher than the outside environment and they were much more gene toxic which could cause more healthy problems to public (Kunzli, Kaiser et al. 2000, Guo, Hu et al. 2014). The underground portion of the subway system is a semi-confined environment that may accumulate either internally generated contaminants or those from the outside environment. Meanwhile, another research indicated that indoor sources can also contribute extremely high levels of outdoor air pollution in heavily populated locations ([14]). Therefore, following by [14], we believe that this subway-induced PM pollution can contribute a lot for outside environment. As suggested by the reviewer, lines 55-56, 495-496 in revised version have been stressed indoor PM pollution is caused by subway systems can create high levels of outside pollution (See lines 55-56, 495).
Point 6: why weather variables are unavailable? The authors use them in the following. (as shown on line 59)
Response 6: After examining the reviewer’s comments carefully, we try to explain it for the following reasons.
First, according to meteorological theory, meteorological conditions restrict the dilution, diffusion, transportation and transformation of atmospheric pollutants, and thus affect the distribution and concentration of atmospheric pollutants. When we estimate the impact of the subway openings on PM concentration, we should match the processing group and the control group first based on the meteorological conditions in order to obtain an unbiased estimate. Therefore, these weather and wind data of both the processing and control groups are very important and necessary for obtaining our final results.
Second, the data of weather variables and wind condition was from two key websites: https://www.wunderground.com/history/daily/ZSOF/date/2017-1-1 and http://data.sheshiyuanyi.com/WeatherData/. However, the latter website only provides the weather and wind data for the city which has a weather station, and the former provides the meteorological data for the relatively large city.
Therefore, we obtained the weather and wind data of 20 cities in the control group and 7cities in the processing group at last. Even though the limited weather and wind data in the control group was available, as yet our findings in this paper are robust by using PM2.5 concentration as a robustness pollutant.
In addition, we must admit that we have not expressed our meaning correctly in the previous manuscript. We have deleted this sentence (See line 61).
Point 7: ‘poor air pollution’ is not clear, please clarify (as shown on line 85)
Response 7: After examining the reviewer’s comments carefully, we must admit that we have not made a clear expression in the previous manuscript. In the revised version, we have made a change (See line 86).
Point 8: again, ‘negative’ is not clear in this context (as shown on line 92)
Response 8: We have re-written this part according to your suggestion (See line 93).
Point 9: ‘healthy’ (as shown on line 97)
Response 9: As suggested by the reviewer, we have made an improvement of this sentence (See line 98).
Point 10: ‘percent’ => % (as shown on lines 119 and 131)
Response 10: As suggested by the reviewer, we have re-written these sentences in the revised paper (See lines 123 and 138).
Point 11: ‘it is support that…’ is not clear, please clarify (as shown on line 150)
Response 11: As suggested by the reviewer, we have re-written the sentence in the revised paper (See lines 154-155).
Point 12: ‘equals to 1’ => ‘that is equal to’ or ‘that equals 1’ (as shown on line 200)
Response 12: As suggested by the reviewer, we have re-written the sentence in the revised paper (See lines 206-209).
Point 13: The statement does not make sense, please rephrase it. (as shown on line 348)
Response 13: As suggested by the reviewer, we have re-written this sentence in the revised paper (See lines 343-347).
Point 14: ‘provide’ => provides (as shown on line 484)
Response 14: As suggested by the reviewer, we have corrected this grammar mistakes in the revised paper (See line 481).
Point 15: ‘another the surprising’ => another surprising (as shown on line 534)
Response 15: As suggested by the reviewer, we have re-written the sentence in the revised paper (See line 533).
Once again, thank you very much for your comments and suggestions.
We tried our best to improve the manuscript and made some changes in the manuscript according to the reviewers’ comments. These changes will not influence the content and framework of the paper.
We appreciate for Editors/Reviewers’ warm work earnestly, and hope that the correction will meet with approval.
Once again, thank you very much for your comments and suggestions.
We look forward to hearing from you in due time regarding our submission and to respond to any further questions and comments you may have.
Yours Sincerely,
Corresponding Author

Reviewer 3 Report
Effective public health policy needs translating scientific research into policy and practice. Providing information about the associations between developing the subway systems and the mitigation of traffic congestion, air pollution and health impact seems very important from the public health point of view.
Presented research article: Can the new subway line openings mitigate PM10 concentration?-Evidence from Chinese cities based on the PSM-DID method provides such evidence-based information, that’s why it is recommended to be publish in International Journal of Environmental Research and Public Health.
The aim of the paper was to estimate the air pollution effect after a new subway line opening. The manuscript focuses on exploring both short- and loner-term effects of the new subway openings on PM10 pollution at the city level in China. Authors also suggest policy implications. Authors used PSM-DID method (that differs from other published approaches) to investigate the effect of the subway line openings on PM10 pollution with adequate robustness checks. Authors also investigated the relationship between the subway line openings and PM10 pollution for cities with varying time windows, population sizes, and levels of economic development in China.
The article meets the relevant quality standards and the main research assumptions made by the Authors are correct. The title adequately inform of its content. Abstract outlines three key elements relevant to the completed
work, sufficiently inform the content of the study. Introduction provide sufficient information about the background and the purpose of the study.
Applied methodology (PSM-DID method) ensures the reliability and validity of the results. Description of the methodology includes all stages of work. As Authors explained the final effect on PM10 pollution after the new subway line openings was ambiguous because even though the decreasing traffic diversion force reduces prolonging travel time on the road and thus the mitigation of PM10 pollution, as yet this mitigated force may be offset by the PM pollutants caused by the subway line supply, which is likely to increase PM10 pollution.
The correct statistical interpretation of the results. The PSM-DID method provided a unique opportunity to be able to avoid the occurrence of sample selectivity bias and individual heterogeneity, and the credibility of the results in the paper was improved by robustness check.
Besides, some results are presented in 10 tables and 2 figures, anyway in legible way. Referencing correct, contains 62 items. The paper is satisfactory in terms of clarity and organization, linguistic correctness and terminological consistency is also an advantage.
It is very important that Authors highlighted limitations of this study. In conclusion the Authors indicated that there is limited evidence regarding of the short- and longer-term relationships between the new subway line openings and air pollution at the city level in China. Furthermore, the investigation on the relationship between a new subway line opening and air pollution usually used RD and DID methods rather than PSM-DID method. The current subway construction in China contributes to improve air quality in terms of PM10 and PM2.5 concentrations in the longer term. Moreover, public transport infrastructure investment should be directed towards cities with over 2 million population sizes in order to reduce PM pollution. However, these reductions in PM may be little due to the subway-induced PM pollution.The manuscript provides harmonized and evidence-based information, which could serve policy-makers and the public.
Author Response
Dear Reviewer 3:
Thank you very much for your comments. We tried our best to improve the manuscript and made some changes in the manuscript according to the reviewers’ comments. These changes will not influence the content and framework of the paper. We appreciate for Editors/Reviewers’ warm work earnestly, and hope that the correction will meet with approval.
Once again, thank you so much.
We look forward to hearing from you in due time regarding our submission and to respond to any further questions and comments you may have.
Yours Sincerely,
Corresponding Author
Round 2
Reviewer 2 Report
I am grateful to the authors for taking into account my observations. Though I still have doubts on the effective role of underground transportation as importante source of PM, the work is well done and future studies will likely solve this problem. It is not my role to challenge the working hypothesis.
This manuscript is a resubmission of an earlier submission. The following is a list of the peer review reports and author responses from that submission.
Round 1
Reviewer 1 Report
I appreciate the authors research. I found a very interesting approach to understanding the implications of public transportation on the effect of PM10 and other types of pollution. I think the methods used are appropriate, well discussed and correct.
I however have a couple of questions regarding your research. You mentioned one study by Moreno, T.; Kelly, F. J.; Dunster, C.; et al. Oxidative potential of subway PM2. 5. Atmospheric Environment 542 2017, 148, 230-238. Is it the only study available? I raised the question because even if subways emit pollution in brakes, friction and such, I am unsure if the amount will outset the amount of pollution that the "new users" of the subway will have.
Furthermore, how would you treat the case in which the "new users" are already using other type of transportation.
I would like to see a table with an estimation of the substitution of transportation, from say automobiles to subway and from other public transportation schemes to subway. Does that varies from city to city.
Another question I have is the relevance of a subway line in a large city. Could you somehow discuss and ideally represent in the model the amount of people moved by the subway.
I am saying so because as I see it, if the subway line is negligible in terms of the amount moved in the city and the substitution of other types of transportation, then it really should not reduce the amount of pollution.
Also, what about the industrial activity of the cities involved. Does GDP captures such situation? I think you should use only the industrial GDP or some other sort of measures including utility plants and so on.
My worry with your model is that the effect that could be seen is only a time effect.
Can you also discuss the alpha2 and the constant? you discuss only alpha1, I bet there is something to say in that regard.
Again, I found it interesting your research but I believe it could be improved.
Best of lucks,
Reviewer 2 Report
This topic points to the very interesting issue of new subway line openings and air quality in China. The research showed the impact of the new subway line openings on PM10 pollution at the city level from 2014 to 2017. There are some specific comments:
What is the difference between your study compared to the previous literature (such as literature [19], [20])? A great deal of work has concentrated on the impact of subway openings on air quality (such as PM2.5, PM10, CO, SO2) in Chinese cities or other regions. In other words, what about your main innovation? The contribution should be clearly presented. In addition, you mentioned the health impact in the abstract, which seems very significant in China, but it was not discussed in the following sections.
Literature review should be extended. The literature basis (now only 35), take 50 as a low threshold. There are several works regarding the impact of subway openings on PM10, PM2.5 concentrations, and the health effect. The literature basis and research basis could be expanded considerably and systematically.
Were there only 7 cities that opened new-built subway lines from 2014 to 2017? Did the new-built mean that the subway line opening was the first time for that city? And in Table 3, you showed province rather than city. You should introduce the sample more clearly and exactly. Why only 20 cities were chosen as the control group?
Discussion of results has to be extended. The reasons for these results should be clarified, and you may analyze the mechanism and test the path. Some more empirical evidence could support the existing results better. What other methods are feasible?How other studies’ results are connected to your results, and what about the reasons for difference?
Good luck for improvement!
Reviewer 3 Report
Journal: International Journal of Environmental Research and Public Health
Title: Can the new subway line openings mitigate PM10 concentration? -Evidence from Chinese cities based on the PSM-DID method
Manuscript Number: IJERPH-770765
Suggested Status: Reject
Comments on IJERPH-770765:
By using PSM-DID method this study tries to investigate the effect of the subway line openings on PM10 pollution at the city level in China. The topic is interesting yet challenging. I appreciate the efforts the authors have made in addressing the issues with this manuscript, however, due to the following concerns, I think the manuscript is not ready for the publication in International Journal of Environmental Research and Public Health:
- The formation of particulate pollution is very complex, highly influenced by meteorology, emissions, secondary formation, etc. This study oversimplified the relationship between related factors especially meteorological variables and particle pollution, and ignored the role that emissions and secondary formation play in PM pollution formation.
- As a background, measures such as emission reduction and urban traffic restrictions have been taken across China in recent years, as a part of the city transportation system, the contribution of subway opening on PM pollution reduction may be weak. Without accurate transportation emission inventory, the relationship between the subway and the reduced emission, as well as PM10 pollution is considered cannot be strictly build up.
I thereby think the data and methodology of this study are not sufficient in sensitivity to serve the research project to detect the effect of the subway line openings on PM10 pollution.
Minor review:
- L28, L194-195: Please provide the reference for the PM10 definition “diameter between 2.5 and 10 μm”. In general, PM10 is defined as particulate matter with a diameter less than 10 μm.
- L189-L191: The statement is vague and hard to understand: “average temperature and relative humidity tend to result in more PM10 pollution, while average wind speed and rain would lead to better PM10 environment”. How “average” meteorological factors (temperature, relative humidity (RH)) corresponding to a more severe PM10 pollution, and “average” wind speed and rain lead to less PM10 pollution?
- L269-270: Please clarify how temperature and RH intensified PM pollution.